# Is Polymicrobial Bacteremia an Independent Risk Factor for Mortality in *Acinetobacter baumannii* Bacteremia?

**DOI:** 10.3390/jcm9010153

**Published:** 2020-01-06

**Authors:** Yung-Chih Wang, Wen-Wei Ku, Ya-Sung Yang, Chih-Chun Kao, Fang-Yu Kang, Shu-Chen Kuo, Chun-Hsiang Chiu, Te-Li Chen, Fu-Der Wang, Yi-Tzu Lee

**Affiliations:** 1Division of Infectious Diseases and Tropical Medicine, Department of Internal Medicine, Tri-Service General Hospital, National Defense Medical Center, Taipei 11490, Taiwan; wystwyst@gmail.com (Y.-C.W.); ysyoung4097@gmail.com (Y.-S.Y.); pipi10279@gmail.com (C.-H.C.); 2Division of Infectious Diseases, Department of Medicine, Taipei City Hospital Renai Branch, Taipei 10629, Taiwan; DAN77@tpech.gov.tw; 3Department of Emergency Medicine, Taipei Veterans General Hospital, Taipei 11217, Taiwan; rosty.kao@gmail.com (C.-C.K.); lynnkang8043@hotmail.com (F.-Y.K.); 4National Institute of Infectious Diseases and Vaccinology, National Health Research Institute, Maoli County 35053, Taiwan; ludwigvantw@gmail.com; 5Graduate Institute of Life Sciences, National Defense Medical Center, Taipei 11490, Taiwan; tecklayyy@gmail.com; 6Division of Infectious Diseases, Taipei Veterans General Hospital, Taipei 11217, Taiwan; fdwang@vghtpe.gov.tw; 7Faculty of Medicine, School of Medicine, National Yang-Ming University, Taipei 11221, Taiwan

**Keywords:** *Acinetobacter baumannii*, appropriate therapy, bacteremia, mortality, polymicrobial infection

## Abstract

This retrospective observational study assessed the differences between monomicrobial and polymicrobial *A. baumannii* bacteremia and identified possible independent risk factors for 14-day mortality. There were 379 patients with *A. baumannii* bacteremia admitted to a tertiary care center in northern Taiwan between August 2008 and July 2015 enrolled for data analysis. Among them, 89 patients (23.5%) had polymicrobial bacteremia and 290 patients (76.5%) had monomicrobial bacteremia. No significant difference in 14-day mortality was observed between patients with monomicrobial and polymicrobial *A. baumannii* bacteremia (26.9% vs. 29.2%, *p* = 0.77). Logistic regression controlled for confounders demonstrated that polymicrobial bacteremia was not an independent predictor of mortality, whereas appropriate antimicrobial therapy was independently associated with reduced mortality. Higher 14-day mortality rates were observed in the polymicrobial bacteremic patients with concomitant isolation of *Escherichia coli*, *Pseudomonas aeruginosa*, and *Enterobacter* spp. from the bloodstream. Compared with patients with monomicrobial multidrug-resistant *A. baumannii* (MDRAb) bacteremia, those with MDRAb concomitant with Gram-negative bacilli bacteremia had a worse outcome. Polymicrobial *A. baumannii* bacteremia was not associated with a higher 14-day mortality rate than that of monomicrobial *A. baumannii* bacteremia, although more deaths were observed when certain Gram-negative bacteria were concomitantly isolated. Appropriate antimicrobial therapy remains an important life-saving measure for *A. baumannii* bacteremic patients.

## 1. Introduction

Polymicrobial bacteremia occurs in 6%–27.4% of patients with bloodstream infections and was first identified as an important problem in the 1960s for its higher attributable mortality rate compared with that for monomicrobial bacteremia [1,2,3,4]. The mortality rate of patients with polymicrobial bacteremia varied from 36% to 63% [1,2,3,4]. This diverse mortality rate may be associated with heterogeneity in the comorbid conditions of the patients [2,5,6], varied infection sources [7], and different causative pathogens [2,3,5,6]. 

*Acinetobacter baumannii* is an emergent nosocomial pathogen with high morbidity and mortality rates [8,9,10]. Bacteremia caused by *A. baumannii* occurs as polymicrobial infections in 19%–35% of cases [6,9,11,12]. Our previous study found that carbapenem-resistant *A. baumannii* (CRAb) provided a sheltering effect for carbapenem-susceptible pathogens and enhanced the pathogenesis of polymicrobial infection [13]. However, no study has evaluated the impact of polymicrobial *A. baumannii* bacteremia on clinical outcomes. Therefore, this retrospective observational study assessed whether polymicrobial infection was an independent risk factor for 14-day mortality in patients with *A. baumannii* bacteremia. 

## 2. Materials and Methods

### 2.1. Study Population

This retrospective study was conducted at Taipei Veterans General Hospital (T-VGH), a 2900-bed tertiary-care teaching hospital located in Taipei, Taiwan, during a seven-year period from August 2008 to July 2015. Charts were reviewed for all patients with at least one blood culture positive for *Acinetobacter calcoaceticus*–*Acinetobacter baumannii* (Acb) complex accompanied by two or more of the following symptoms and signs of infection: fever, hypothermia, tachypnea, tachycardia, leukocytosis, or leukopenia. We included only the first blood culture for patients with two or more positive blood cultures. Patients under 20 years of age and those with incomplete medical records were excluded. The protocol was approved by the institutional review boards (IRBs) of T-VGH (IRB No. 2016-05-019CC). 

### 2.2. Data Collection

We reviewed the medical records to retrieve clinical information, including demographic characteristics, comorbidities, duration of intensive care unit (ICU) and hospital stays, time of receipt, dose and route of therapy with individual antimicrobial drugs, receipt of invasive procedures, time of bacteremia onset, and infectious foci. The onset of bacteremia was defined as the day when the blood culture that eventually yielded *A. baumannii* was drawn. Polymicrobial bacteremia was defined as the simultaneous isolation of *A. baumannii* and one or more microorganisms from the blood during the same bacteremic episode. However, *Corynebacterium* spp., *Bacillus* spp., *Propionibacterium acnes*, and skin flora (e.g., *Micrococcus* spp., *Streptococcus viridans*, and coagulase-negative *staphylococci*) were considered contaminants and not included in the analyses unless they were related to device infection and grown in two or more blood cultures [6,14]. Episodes of bloodstream infection were defined as ICU-acquired bacteremia if they appeared 48 h after ICU admission. Chronic kidney disease was defined as an estimated glomerular filtration rate <60 mL/min/1.73 m^2^. Neutropenia was defined as an absolute neutrophil count <0.5 × 10^9^ neutrophils/L. Immunosuppressive therapy was defined as the receipt of cytotoxic agents within six weeks [15], corticosteroids at a dosage equivalent to or higher than 15 mg of prednisolone daily for at least seven days within four weeks, or other immunosuppressive agents within two weeks before bacteremia onset [15,16]. A history of recent surgery was defined as receipt of operations within four weeks before bacteremia onset. The severity of infection was evaluated using the Acute Physiology and Chronic Health Evaluation (APACHE) II score [17] within 24 h before bacteremia onset. The primary infection source of bacteremia was determined according to the definitions from the Centers for Disease Control and Prevention [18]. Appropriate antimicrobial therapy was defined as the administration of at least one antimicrobial agent to which the causative pathogen was susceptible in vitro within 48 h after the onset of bacteremia, with an approved route and dosage appropriate for end organ(s) function. In patients with polymicrobial bacteremia, a combination of antimicrobial agents covering all isolated microbes was deemed appropriate. Monotherapy with an aminoglycoside was not considered appropriate therapy for *A. baumannii* bacteremia. The primary outcome measure was all-cause 14-day mortality following bacteremia onset. Fourteen-day mortality was chosen as the endpoint to allow adequate ascertainment of treatment response. We reasoned that 30 days is too long for critically ill patients, as there are many competing causes of death, and seven days is too short a time to witness a response to treatment.

### 2.3. Microbiological Studies

Bacteria were phenotypically identified to Acb complex using a Vitek 2 system (bioMérieux, Marcy l’Etoile, France). Other bacterial species were also identified using this system. All isolates were regrown from –80 °C storage, identified to the species level, and tested for their susceptibility to various antimicrobials. A multiplex polymerase chain reaction (PCR) method was used to identify *A. baumannii* to the genomic species level [19]. Antimicrobial susceptibilities of *A. baumannii* and other concomitant isolated bacterial pathogens were determined by the agar dilution method [20] and interpreted according to the Clinical Laboratory Standards Institute (CLSI) criteria [21].

Multidrug resistance was defined as resistance to three or more of the following classes of antimicrobials: anti-pseudomonal cephalosporins, anti-pseudomonal carbapenems, ampicillin/sulbactam, fluoroquinolones, and aminoglycosides. Carbapenem resistance was defined as resistance to imipenem, meropenem, or doripenem. 

### 2.4. Statistical Analysis

Chi-square tests with Yates correction or Fisher’s exact tests were used to compare categorical variables; Student’s t or Mann–Whitney rank-sum tests were used to analyze continuous variables, as appropriate. Logistic regression models were used to explore independent risk factors of 14-day mortality. We performed univariate analyses separately for each risk factor to ascertain the odds ratios (ORs) and 95% confidence intervals (CIs). All biologically plausible variables with a *p* <0.20 in the univariate analysis were considered for inclusion in the logistic regression model in the multivariable analysis. A backward selection process was utilized. Interactions between APACHE II score and the covariates were also examined in the logistic regression model. Time to mortality, defined as the interval between bacteremia onset and death, was analyzed by Kaplan–Meier survival analysis with log-rank tests. *p* < 0.05 was considered statistically significant. SPSS Statistics for Windows, version 22.0 (IBM Corp., Armonk, NY, USA) was used for all data analyses.

## 3. Results

During the seven-year study period, we enrolled 732 patients with at least one set of blood cultures positive for Acb complex. A total of 379 patients met the inclusion criteria, of which 89 (23.5%) patients had polymicrobial *A. baumannii* bacteremia. The demographic and clinical characteristics of patients with polymicrobial and monomicrobial *A. baumannii* bacteremia are summarized in Table 1. Patients with polymicrobial *A. baumannii* bacteremia were older, more likely to have coronary artery disease, less frequently experienced shock within three days of bacteremia onset, and more frequently received inappropriate antimicrobial therapy (13.5% vs. 38.6%, *p* < 0.001), compared with those in patients with monomicrobial bacteremia. No significant difference in 14-day mortality (29.2% vs. 26.9%, *p* = 0.770) and 30-day mortality (38.2% vs. 34.8%, *p* = 0.561) was observed between these two groups. To analyze the duration of in-hospital stay after the onset of bacteremia, we separated the population into two groups, survivors and non-survivors, at hospital discharge. In the patients who survived to discharge, those with polymicrobial *A. baumanii* bacteremia and monomicrobial *A. baumanii* bacteremia had a comparable hospital stay after the onset of bacteremia (21.9 [9–35] vs. 21.9 [9–34] days, *p* = 0.997). In terms of the patients who died in hospital, those with polymicrobial *A. baumanii* bacteremia and monomicrobial *A. baumanii* bacteremia also had a comparable hospital stay after the onset of bacteremia (13.9 [0.75–26.5] vs. 13.7 [1–21] days, *p* = 0.944). The 30-day re-admission rates of survived patients with polymicrobial and monomicrobial *A. baumannii* bacteremia were not significantly different (17.9% and 13.2%, respectively; *p* = 0.620).

Logistic regression analysis was performed to determine the risk factors for 14-day mortality in the entire population with *A. baumannii* bacteremia, as shown in Table 2. Polymicrobial bacteremia was not independently associated with increased 14-day mortality. Malignancy, shock, and higher APACHE II score were independent predictors of 14-day mortality, whereas appropriate antimicrobial therapy was independently associated with reduced 14-day mortality (*p* = 0.001). The predictors for 30-day mortality among patients with *A. baumannii* bacteremia were analyzed in the same fashion as the above analyses. The results were similar to those for prediction of 14-day mortality (Appendix A).

We stratified the 89 patients with polymicrobial *A. baumannii* bacteremia according to the concomitantly isolated bacterial species (Table 3). Coagulase-negative staphylococci were the most common isolated microorganisms (20 cases). Among them, 17 patients (85.0%) were caused by a device-related infection. Nine patients (52.9%) had their infected device removed at a median duration of six days (interquartile range, 4–8) after the onset of the bacteremia. However, the removal of device did not affect their clinical outcome. There were 12 patients with polymicrobial *A. baumannii* bacteremia that received appropriate antimicrobial therapy. Half of them received combination therapy. Carbapenem plus vancomycin/teicoplanin was the most frequently used antimicrobial regimen for those with polymicrobial *A. baumannii* bacteremia (Appendix A).

Although no significant difference in 14-day mortality rates was observed between patients with polymicrobial and monomicrobial *A. baumannii* bacteremia, the rate in polymicrobial *A. baumannii* bacteremia varied according to the concomitant pathogen. Concomitant *Escherichia coli* bacteremia had the highest 14-day mortality rate (71.4%), followed by *Pseudomonas aeruginosa* (50.0%) and *Enterobacter* spp. bacteremia (37.5%). The 14-day mortality rate was higher in patients who had polymicrobial *A. baumannii* bacteremia with concomitant Gram-negative bacilli (GNB) than in patients with concomitant Gram-positive cocci (GPC) (Table 3; 38.9% vs. 19.1%, *p* = 0.053) (Figure 1; *p* = 0.038, log-rank test). The rates of methicillin-resistant *Staphylococcus aureus* (MRSA), extended-spectrum beta-lactamase (ESBL)-producing *K. pneumoniae*, ESBL-producing *E. coli*, and AmpC-producing *E. coli* were 80.0%, 16.7%, 14.3%, and 14.3%, respectively (Appendix A). The presence of these resistant bacteria was not associated with significantly higher 14-day mortality.

To investigate the influence of *A. baumannii* antibiotic resistance on the mortality of patients with polymicrobial bacteremia with concomitant GNB infection, we extracted data of those with multidrug-resistant *A. baumannii* (MDRAb) or carbapenem-resistant *A. baumannii* (CRAb) infections for further analysis. Patients with MDRAb and concomitant GNB bacteremia had a higher 14-day mortality rate than patients with monomicrobial MDRAb bacteremia (Figure 2; *p* = 0.036, log-rank test). Patients with CRAb and concomitant GNB bacteremia had a higher 14-day mortality rate than patients with monomicrobial CRAb bacteremia (50.0% vs. 30.6%, *p* = 0.285). The difference was not statistically significant in this small cohort (*n* = 82). 

The survival analysis at 30 days was analyzed in the same fashion as the above analyses (Appendix A). The results were similar to those for 14-day mortality.

## 4. Discussion

While polymicrobial *A. baumannii* bacteremia is not uncommon, its clinical significance has not previously been elucidated. This retrospective study assessed the differences between monomicrobial and polymicrobial *A. baumannii* bacteremia and identified possible independent risk factors for 14-day mortality for these infections. We found that polymicrobial bacteremia was not an independent predictor of mortality, whereas appropriate antimicrobial therapy was independently associated with reduced mortality. 

This study, in agreement with previous studies [6,9,11,12], observed that approximately one-fourth of patients with *A. baumannii* bloodstream infections presented with polymicrobial bacteremia. Owing to its high prevalence, investigation of the clinical significance of polymicrobial *A. baumannii* bacteremia is warranted. These patients were older and more likely to have coronary artery disease and experienced shock at the time of bacteremia. Moreover, as few as one out of seven patients received appropriate antimicrobial therapy directed against every microbial species isolated from the blood. As the clinical characteristics differed between monomicrobial and polymicrobial *A. baumannii* bacteremia, studies on the clinical outcomes of patients with monomicrobial and polymicrobial *A. baumannii* bacteremia should evaluate the infections separately. Furthermore, when evaluating the efficacy of various regimens on the clinical outcome of *A. baumannii* bacteremia, polymicrobial *A. baumannii* bacteremia should be excluded, unless the antibiotic efficacy against the concomitant isolated microorganisms was also evaluated.

Our results showed that the 14-day mortality rate was not significantly higher in patients with polymicrobial than in patients with monomicrobial *A. baumannii* bacteremia. This finding is consistent with those of previous studies and suggests that polymicrobial infection was not an independent risk factor for mortality of *A. baumannii* bacteremia [10,11,22,23]. Although several studies have reported higher mortality rates in patients with polymicrobial bacteremia [3,5,6,24], the attributable mortality rate of polymicrobial bacteremia varied among causative pathogens [3,25,26]. Compared with monomicrobial bacteremia, polymicrobial bacteremia involving *Staphylococcus aureus* [27], *Enterococcus* spp. [28], and *P. aeruginosa* [29] have higher mortality rates. In contrast, polymicrobial *Klebsiella pneumoniae* bacteremia did not result in a worse prognosis [30]. These findings indicate that the influence of polymicrobial bacteremia on clinical prognosis should be investigated separately according to the causative pathogens.

The interbacterial interactions in polymicrobial infection, which vary among co-existing species, include metabolite exchange, co-aggregation, interspecies signaling, antibiotic resistance, and biofilm formation [31]. While most co-existing bacteria promote their pathogenesis through these interactions, some of them exhibit a competitive relationship [31]. Thus, these interbacterial interactions in polymicrobial infection may have two-sided effects on the disease severity. Therefore, the different co-existing bacteria in polymicrobial infections may have diverse effects on the clinical outcome. This in line with our findings that different concomitant bacterial pathogens had heterogeneous clinical outcomes. Compared with monomicrobial *A. baumannii* bacteremia, the higher mortality rate observed in patients with concomitant GNB bacteremia and lower mortality rate in those with concomitant GPC bacteremia suggested that the interactions between *A. baumannii* and GNB had negative effects, while those between *A. baumannii* and GPC positively influenced the disease severity. 

On the basis of previous studies showing that beta-lactamase-producing pathogens can provide indirect pathogenesis by protecting the other pathogens in polymicrobial infection environments [13], we assessed whether the co-existence of drug-resistant *A. baumannii* would protect the concomitant GNB and result in a worse clinical outcome. Our findings were consistent with those of previous studies reporting higher 14-day mortality in patients with polymicrobial bacteremia with concomitant MDRAb and GNB than in those with monomicrobial MDRAb bacteremia. Although patients with polymicrobial bacteremia with concomitant CRAb and GNB had a higher 14-day mortality rate than those with monomicrobial CRAb bacteremia, the difference was not statistically significant in this small population. 

Previous studies of monomicrobial *A. baumannii* bacteremia also reported inappropriate antimicrobial therapy to be an independent risk factor for mortality [9,11,12,22,32]. The results of the present study strengthened these findings by enrolling a large number of patients and including a number of cases with polymicrobial *A. baumannii* bacteremia. Ours and previous studies consistently found that inappropriate antimicrobial therapy was significantly associated with survival in patients with acquired polymicrobial bacteremia [2]. When studies evaluating the efficacy of appropriate antimicrobial therapy enroll both monomicrobial and polymicrobial *A. baumannii* bacteremia, appropriate therapy should be defined as a single or a combination of antimicrobial agents covering all isolated microbes.

The weakness of this study was the retrospective design required to enroll sufficient patients for detailed analysis at a single tertiary care medical center. The major strength of our study was its inclusion of a number of patients with genomically-defined *A. baumannii* bacteremia, consistency of patient care in a single center, stringent definition of the appropriateness of antimicrobial therapy, and a well-defined primary endpoint of 14-day mortality. We also conducted multivariate analysis to assess clinical variables that were significantly associated with mortality owing to *A. baumannii* bacteremia. 

## 5. Conclusions

Polymicrobial *A. baumannii* bacteremia was not an independent risk factor for 14-day mortality. Regardless of monomicrobial or polymicrobial infection, appropriate antimicrobial therapy remains an important life-saving measure for *A. baumannii* bacteremia patients.

## Figures and Tables

**Figure 1 jcm-09-00153-f001:**
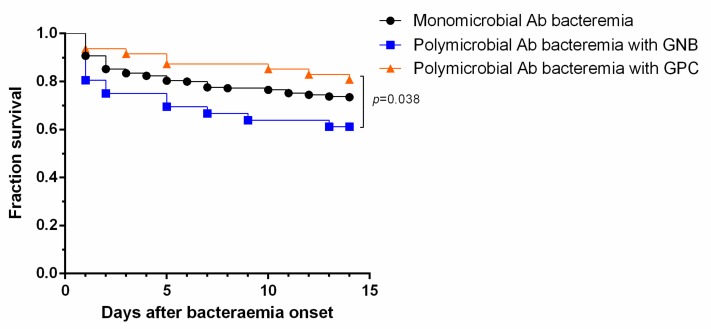
Kaplan–Meier plot showing the survival rates of patients with acquired monomicrobial *Acinetobacter baumannii* (Ab) bacteremia, polymicrobial Ab bacteremia with concomitant Gram-negative bacilli (GNB), and polymicrobial Ab bacteremia with concomitant Gram-positive cocci (GPC) (polymicrobial Ab bacteremia with concomitant GNB versus monomicrobial Ab bacteremia, *p* = 0.098 by log-rank test; polymicrobial Ab bacteremia with concomitant GPC versus monomicrobial Ab bacteremia, *p* = 0.265 by log-rank test).

**Figure 2 jcm-09-00153-f002:**
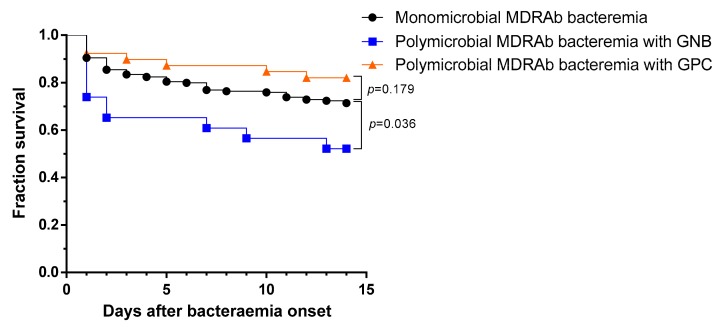
Kaplan–Meier plot showing the survival rates of patients with acquired polymicrobial multidrug-resistant *Acinetobacter baumannii* (MDRAb) bacteremia with concomitant Gram-negative bacilli (GNB), polymicrobial Ab bacteremia with concomitant Gram-positive cocci (GPC), and those with acquired monomicrobial MDRAb bacteremia. *p*-values were calculated by log-rank tests. (polymicrobial MDRAb bacteremia with concomitant GNB versus polymicrobial MDRAb bacteremia with concomitant GPC, *p* = 0.009 by log-rank test).

**Table 1 jcm-09-00153-t001:** Demographic and clinical characteristics of patients with monomicrobial and polymicrobial *Acinetobacter baumannii* bacteremia.

Characteristic	All (*n* = 379)	Polymicrobial (*n* = 89)	Monomicrobial (*n* = 290)	*p*
**Demographic characteristics**				
Age, years	74 (60–81)	78 (64–82.5)	73 (59–81)	0.012
Male sex	280 (73.9)	66 (74.2)	214 (73.8)	1.000
Community-onset	20 (5.3)	2 (2.2)	18 (6.2)	0.181
Acquired in ICU	189 (49.9)	47 (52.8)	142 (49.0)	0.608
Days of hospitalization before bacteremia	18 (5–34)	21 (4.5–48)	16 (5.8–32)	0.161
**Comorbidity**				
Alcoholism	30 (7.9)	9 (10.1)	21 (7.2)	0.514
Liver cirrhosis	23 (6.1)	5 (5.6)	18 (6.2)	1.000
Chronic kidney disease	90 (23.7)	28 (31.5)	62 (21.4)	0.070
Type 2 diabetes mellitus	99 (26.1)	20 (22.5)	79 (27.2)	0.448
Chronic obstructive pulmonary disease	68 (17.9)	17 (19.1)	51 (17.6)	0.867
Hypertension	127 (33.5)	31 (34.8)	96 (33.1)	0.862
Coronary artery disease	56 (14.8)	22 (24.7)	34 (11.7)	0.004
Congestive heart failure	43 (11.3)	13 (14.6)	30 (10.3)	0.359
Cerebral vascular accident	71 (18.7)	17 (19.1)	58 (20.0)	0.973
Collagen vascular disease	18 (4.7)	5 (5.6)	13 (4.5)	0.775
Malignancy	128 (33.8)	28 (31.5)	100 (34.5)	0.690
Neutropenia	18 (4.7)	4 (4.5)	14 (4.8)	1.000
Chemotherapy	39 (10.3)	6 (6.7)	33 (11.4)	0.289
Immunosuppressive therapy	19 (5.0)	4 (4.5)	15 (5.2)	1.000
Recent surgery (within 4 weeks)	126 (33.2)	26 (29.2)	100 (34.5)	0.427
Trauma	11 (2.9)	1 (1.1)	10 (3.4)	0.470
Shock within 3 days	15 (4.0)	0 (0.0)	15 (5.2)	0.027
APACHE II score within 24 h before bacteremia onset	24 (18–31)	24 (18.3–31)	24 (27–31)	0.820
**Invasive procedures ^a^**				
Abdominal drainage	33 (8.7)	3 (3.4)	30 (10.3)	0.068
Arterial catheter	76 (20.1)	15 (16.9)	61 (21.0)	0.478
Central venous catheter	196 (51.7)	51 (57.3)	145 (50.2)	0.291
Foley catheter	221 (58.3)	53 (59.6)	168 (57.9)	0.882
Hemodialysis	27 (7.1)	8 (9.0)	19 (6.6)	0.585
Nasogastric tube	255 (67.3)	63 (70.8)	192 (66.2)	0.499
Pulmonary artery catheter	41 (10.8)	7 (7.9)	34 (11.7)	0.406
Total parental nutrition	27 (7.1)	2 (2.2)	25 (8.6)	0.070
Tracheotomy	46 (12.1)	16 (18.0)	30 (10.3)	0.081
Ventilator	192 (50.7)	51 (57.3)	141 (48.6)	0.190
**Infection source ^b^**				
Respiratory tract	213 (56.2)	54 (60.7)	159 (54.8)	0.395
Urinary tract	31 (8.2)	4 (4.5)	27 (9.3)	0.219
Catheter-related	35 (9.2)	11 (12.4)	24 (8.3)	0.340
Intra-abdomen	26 (6.9)	3 (3.4)	23 (7.9)	0.212
Soft tissue or wound	12 (3.2)	4 (4.5)	8 (2.8)	0.487
Primary bacteremia	71 (18.7)	16 (18.0)	55 (19.0)	0.957
**Resistance profiles of the bloodstream isolate**				
Multidrug resistance	265 (69.9)	66 (74.2)	199 (68.6)	0.387
Carbapenem resistance	91 (24.0)	19 (21.3)	72 (24.8)	0.596
Appropriate antimicrobial therapy	124 (32.7)	12 (13.5)	112 (38.6)	<0.001
14-day mortality	104 (27.4)	26 (29.2)	78 (26.9)	0.770
30-day mortality	135 (35.6)	34 (38.2)	101 (34.8)	0.561

NOTE. Data are median values (interquartile range) for continuous variables and numbers of cases (%) for categorical variables. ICU, intensive care unit; APACHE II, Acute Physiologic and Chronic Health Evaluation II. ^a^ At the time that blood culture was obtained. ^b^ Patients may have more than one source of bacteremia.

**Table 2 jcm-09-00153-t002:** Logistic regression analysis of predictors for 14-day mortality among patients with *Acinetobacter baumannii* bacteremia.

Demographic or Characteristic	Univariable Analysis		Multivariable Analysis	
	Odds Ratio (95% CI)	*p*	Odds Ratio (95% CI)	*p*
Chronic kidney disease	1.617 (0.946–2.765)	0.079		
Coronary artery disease	1.735 (0.956–3.148)	0.070		
Malignancy	1.667 (1.046–2.657)	0.032	2.554 (1.373–4.753)	0.003
Neutropenia	3.551 (1.361–9.263)	0.010		
Recent surgery (within 4 weeks)	0.410 (0.239–0.701)	0.001		
Shock within 3 days	4.081 (2.386–6.981)	<0.001	2.430 (1.220–4.843)	0.012
APACHE II score	1.169 (1.129–1.210)	<0.001	1.173 (1.130–1.216)	<0.001
Arterial catheter	2.170 (1.280–3.680)	0.004		
Central venous catheter	1.682 (1.060–2.668)	0.027		
Foley catheter	2.285 (1.400–3.729)	0.001		
Nasogastric tube	2.198 (1.293–3.735)	0.004		
Ventilator	2.054 (1.291–3.267)	0.002		
Respiratory tract infection	2.015 (1.252–3.243)	0.004		
Urinary tract infection	0.262 (0.078–0.881)	0.030		
Appropriate antimicrobial therapy	0.525 (0.313–0.881)	0.015	0.317 (0.162–0.621)	0.001
Polymicrobial bacteremia	1.122 (0.663–1.897)	0.668		

Abbreviations: ICU, intensive care unit; APACHE II, Acute Physiologic and Chronic Health Evaluation II; CI, confidence interval.

**Table 3 jcm-09-00153-t003:** Characteristics of patients with polymicrobial *Acinetobacter baumannii* bacteremia stratified by concomitantly isolated bacterial species.

Concomitantly Isolated Bacterial Species	No. (%) of Patients	APACHE II Score, Mean (Interquartile Range)	No. (%) of Patients		
			Appropriate Antimicrobial Therapy	Concomitant MDR/CR *A. baumannii*	14-Day Non-Survival
Gram-positive cocci (GPC)	47 (52.8)	23 (18–28)	5 (10.6)	39/8 (83.0/17.0)	9 (19.1)
Coagulase-negative staphylococci (CNS)	20 (22.5)	18.5 (11.25–25.75)	1 (5.0)	15/3 (78.0/15.0)	2 (10.0)
*Enterococcus* spp.	12 (13.5)	24.5 (18.5–37.75)	2 (16.7)	10/0 (83.3/0)	3 (25.0)
*Staphylococcus aureus*	4 (4.5)	24.5 (22–35.25)	1 (25.0)	4/2 (100.0/50.0)	1 (25.0)
*Enterococcus* spp. + *S. aureus*	4 (4.5)	26.5 (9.25–28.75)	0 (0)	3/1 (75.0/25.0)	0 (0)
CNS + *S. aureus*	2 (2.3)	19, 24	0 (0)	1/1 (50.0/50.0)	0 (0)
Other GPC^a^	5 (5.6)	25 (22.5–30.5)	1 (20.0)	5/1 (100.0/20.0)	3 (60.0)
Gram-negative bacilli (GNB)	36 (40.5)	24 (19–35)	6 (16.7)	23/10 (63.9/27.8)	14 (38.9)
*Pseudomonas aeruginosa*	8 (9.0)	33.5 (18.25–38.75)	2 (25.0)	6/2 (75.0/25.0)	4 (50.0)
*Enterobacter spp.*	8 (9.0)	26 (22.25–40)	1 (12.5)	5/2 (62.5/25.0)	3 (37.5)
*Escherichia coli*	7 (7.9)	31 (24–39)	1 (14.3)	6/2 (85.7/28.6)	5 (71.4)
*Klebsiella pneumoniae*	6 (6.7)	20 (17.5–31.5)	0 (0)	3/1 (50.0/16.7)	1 (16.7)
Other GNB^b^	7 (7.9)	18 (10–27)	2 (28.6)	4/4 (42.9/42.9)	1 (14.3)
Yeast^c^	3 (3.4)	18, 23, 33	1 (33.3)	2/1 (66.7/33.3)	1 (33.3)
Mixed GNB and GPC	3 (3.4)	19, 24, 47	0 (0)	2/0 (66.7/0)	2 (66.7)

Abbreviations: APACHE II, Acute Physiologic and Chronic Health Evaluation II; MDR, multidrug resistance; CR, carbapenem resistance. ^a^ Includes *Aerococcus* (*n* = 1), concomitant Coagulase-negative staphylococci and *Enterococcus spp.* (*n* = 1)*, Lactobacillus spp.* (*n* = 1)*, Listeria* (*n* = 1)*,* and *Streptococcus spp*. (*n* = 1). ^b^ Includes *Aeromonas sobria* (*n* = 1), *Alcaligenes xylosoxidans* (*n* = 1)*, Chryseobacterium indologens* (*n* = 1), *Citrobacter diversus* (*n* = 1)*, Pantoea agglomerans* (*n* = 1)*, Proteus mirabilis* (*n* = 1), and *Serratia marcescens* (*n* = 1). ^c^ Includes *Candida glabrata* (*n* = 1)*; Candida tropicalis* (*n* = 1)*;* and concomitant coagulase-negative staphylococci, *Staphylococcus aureus*, and *Candida albicans* (*n* = 1).

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
