# Peer review of "Is Polymicrobial Bacteremia an Independent Risk Factor for Mortality in Acinetobacter baumannii Bacteremia?"

_jcm, 2020, doi:10.3390/jcm9010153_

Round 1
Reviewer 1 Report
General comments The authors conducted a single-center, retrospective study in a large tertiary hospital in Taiwan to investigate the differences between monomicrobial and polymicrobial A. baumanii bacteremia and identify possible independent risk factors for 14-days mortality. They found 379 patients with mono- or poly-microbial A. baumanii bacteremia. Although 14-days mortality between mono- and poly- A. baumanii was not significantly different, appropriate antimicrobial therapy was independently associated with a favorable outcome. Overall, the manuscript was well written and the contents of the manuscript were interesting for clinicians.
Specific recommendation for revision
1.Primary outcome: 14-days mortality
The authors used 14-days mortality as the primary outcome in the current study. However, several severe patients may not be finished at 14-days after the onset of infection. Thus, I believe it is useful to investigate 30-days mortality as a secondary outcome. Could you add the analysis for 30-days mortality and add the result in Table 1? Also, it may be interesting to assess the duration of hospitalization, duration of bacteremia (how many days the bacteremia persists?), duration of fever in each group.
2.Page 4, line 141: concomitant GNB/GPC bacteremia
The rate of resistant GNB/GPC organisms may influence mortality. Please show the number or rate of resistant GNB/GPC such as MRSA, ESBL, AmpC, metallo-beta-lactamases producing organisms.
3. Page 4, Figure 2 Please add the line for Polymicrobial MDRAb bacteremia with GPC the same as Figure 1.
Reviewer 2 Report
The authors performed a retrospective review to determine the clinical characteristics and impact of polymicrobial Acinetobacter baumanii bacteremia on outcome. Results indicated that the most significant predictor of mortality was inappropriate antibiotic therapy. However, details regarding antibiotic therapy were missing such as agents prescribed for empiric vs directed therapy, monotherapy vs combination, time to receipt of appropriate therapy. Providing details on antibiotic management could strengthen the conclusion of this paper.
Materials and methods, study population (lines 63-64): The authors mention that patients with positive cultures accompanied by symptoms and signs of infection were included in the study. It would be helpful to know what variables, subjective and objective, were screened for infection.
Related to first bullet point, were there cases that were excluded because bacteremia did not appear to be a true infection?
Coagulase-negative staphylococci were reported to be the most common isolated microorganisms (lines 136-137). Inclusion of such organisms were deemed to be not a contaminant presumably due to it being a device related infection. What proportion of patients had device-related infection? And what % had their infected device removed ? time to removal? And how does device management affect their outcome ?
The authors mention that appropriate antimicrobial therapy was independently associated with reduced mortality (lines 184-186): were there certain combinations of appropriate antimicrobial agents that predominated in polymicrobial bacteremia? agents prescribed for empiric vs directed therapy, monotherapy vs combination, time to receipt of appropriate therapy
This study reports that polymicrobial baumannii bacteremia is not an independent risk factor for 14-day mortality. What was the rationale for choosing all-cause 14-day mortality? What about looking at 14-day mortality related to infection? Moreover, did the investigators look into other outcomes such as 30-day mortality, readmission rates, time to clinical stability, etc.?
Reviewer 3 Report
The manuscript is well written, designed, and important to clinicians since Acinetobacter baumanii bacteremia is a common hospital acquired infection in many hospitals worldwide.
I would add to the abstract the number of patients in each group.
In the methods, line 83-83 - not clear why 6 weeks of immunosuppressive therapy was defined, and not 3 months (as defined for vaccination in this population for example), and why 7 days of prednisone were defined as immunosuppression, while usually 14-21 days at least are considered as risk factor for immunesuppression.
Line 95, all-cause mortality was evaluated for 14 days, I guess 14 days are reasonable for severe infection like acinetobacter baumanii sepsis, still an explanation is warranted since most of the studies define 28-30 days of follow up for mortality.
